# Customized Titanium Mesh for Guided Bone Regeneration with Autologous Bone and Xenograft

**DOI:** 10.3390/ma15186271

**Published:** 2022-09-09

**Authors:** Anna Bertran Faus, José Cordero Bayo, Eugenio Velasco-Ortega, Aina Torrejon-Moya, Francesca Fernández-Velilla, Fernando García, José López-López

**Affiliations:** 1Faculty of Medicine and Health Sciences (Dentistry), University of Barcelona, 08907 L’Hospitalet de Llobregat, Spain; 2Department of Comprehensive Dentistry for Adults and Gerodontology, Faculty of Dentistry, University of Seville, 41018 Seville, Spain; 3Department of Oral Medicine, Faculty of Dentistry, Service of the Medical-Surgical Area of Dentistry Hospital, University of Barcelona, 08907 Barcelona, Spain

**Keywords:** guided bone regeneration, vertical regeneration, customized titanium mesh, patient-specific titanium mesh

## Abstract

The augmentation of the alveolar crest after the loss of one or several teeth can be carried out using different bone augmentation techniques. These techniques include bone distraction, ridge expansion, bone block grafts, etc. Guided bone regeneration is an alternative to increase the volume of the hard tissues for the subsequent placement of the implants in the optimal three-dimensional position. The objective of this paper is to show a case report of the use of customized titanium mesh for posterior vertical bone regeneration. Case report and Results: A 59-year-old woman comes to rehabilitate edentulous spaces with implants. After taking the anamnesis and the intra and extraoral exploration, a vertical and horizontal bone defect is observed in the third quadrant. After the radiological study with CBCT, a bone height of 6.04 mm to the inferior alveolar nerve and a width of the bone crest of 3.95 mm was observed. It was decided to carry out a regeneration with a preformed titanium mesh (Avinent^®^, Santpedor, Spain) and four microscrews (Avinent^®^, Santpedor, Spain). The flap was closed without tension. Regular check-ups were performed without complications. At 7 months, the mesh was removed and two osteoingrated implants (Avinent^®^, Santpedor, Spain) were placed with a torque greater than 45 N/cm and an ISQ of 82 and 57 N/cm, respectively. The bone gain obtained was 1.84 and 1.92 mm in width and 4.2 and 3.78 mm in height for positions 3.5 and 3.6. The newly formed bone, obtained by trephine, was well-structured and histologically indistinguishable from the previous bone. Conclusion: The use of a customized pre-formed titanium mesh together with the mixture of autologous bone and xenograft is a feasible and predictable technique for vertical bone regeneration.

## 1. Introduction

After tooth extraction, a remodeling process occurs in the alveolar bone that results in a horizontal and vertical decrease in the alveolar crest. This resorption can cause inadequate bone volume for the correct insertion of dental implants [1]. The data also revealed that the cure rate varies markedly between subjects [1].

Different bone regeneration techniques aim to increase bone volume for the correct three-dimensional placement of the implant, restore the relationship of the intermaxillary ridge, increase aesthetic results, meet the biomechanical requirements of the prosthesis, and obtain healthy bone to facilitate osseointegration and long-term implant survival and success. Posterior mandibular regeneration techniques include bone grafts (inlay or onlay) [2,3], bone distraction [4], lower dental nerve transposition, and guided bone regeneration, among others [5,6].

In the last 10 years, guided bone regeneration has been shown to be a viable technique for vertical and horizontal regeneration. This technique consists of cellular exclusion utilizing a membrane that acts as a barrier. The application of barrier membranes is a key factor in the success of GBR [7]. This can be done through the use of resorbable membranes (for example, collagen membrane) or non-resorbable membranes (polytetrafluoroethylene membrane, titanium-reinforced PTFE, titanium mesh, etc.) [1,6].

Resorbable barrier membranes, such as collagen membranes, are widely used in the clinic because they have high biocompatibility and do not require a second surgery to remove [8]. However, uncontrolled degradation, insufficient stiffness, and space maintenance often lead to inadequate bone regeneration [9].

To increase bone volume in height, the use of non-resorbable membranes with titanium reinforcement is recommended, since these will allow the space to be maintained for a longer period and prevent the collapse of the area to be regenerated, compared to non-resorbable membranes [10]. Vertically regenerated bone with guided bone regeneration techniques responds to dental implant placement in the same way as non-regenerated native bone [10].

Titanium mesh is widely used for guided bone regeneration due to its high rigidity to provide space maintenance, low density, corrosion resistance, and biocompatibility. The use of these non-resorbable membranes is not without complications; the main complication of these membranes is early or late exposure of the membrane, causing infection of the biomaterial and compromising future regeneration [11]. Today, membrane exposure can be assumed to be a “predictable” complication [11].

On the other hand, a traditional Ti mesh does not conform to the anatomical shape of the area of a given bone defect, and intraoperative cutting and bending of the Ti mesh may increase the risk of postoperative exposure and repeated mucosal irritation [12]. Therefore, preformed, three-dimensional (3D), customized barrier membranes with favorable mechanical properties would be preferable for ideal bone regeneration. Continuous advances in 3D computer-aided planning and design application, as well as computer-aided manufacturing [13] facilitate the fabrication of these barriers in different materials: custom titanium [14], polyether-ether ketone (PEEK) [15] and unsintered hydroxyapatite/poly-l-lactide (uHA/PLLA) meshes [16]. This 3D fabrication allows it to perfectly adapt to the anatomical shapes of the bone defect areas, achieving a precise reconstruction of the lost volume [17]. Recent reviews on these aspects can be found in the works of Roca-Millan et al. [18], Shi et al. [19] or Xie et al. [20].

The objective of this case is to show that the use of a customized preformed rigid titanium mesh is a viable option for posterior vertical bone regeneration if an adequate clinical protocol is followed.

## 2. Materials and Methods

A 59-year-old female patient with no relevant medical history or known allergies attends the Master of Oral Medicine, Surgery and Implantology of the Dental Hospital of the University of Barcelona (UB) to assess the implant-supported rehabilitation of the edentulous space of the third quadrant. Intraoral examination shows the absence of 3.5 and 3.6 together with a vertical bone defect. (Figure 1). In the CBCT, a bone height to the inferior alveolar nerve of 6.04 mm and a width of the crest of 3.95 mm is observed.

After analyzing the case, and based on the need for guided bone regeneration, it was decided to perform it using a personalized non-resorbable membrane technique made of titanium. Specifically, a titanium mesh made of Ti6A14V ELI, without surface treatment, with a thickness of 0.4 mm and a pore size of 2 mm. Its manufacture is carried out based on virtual planning and design with the 3D files of the CBCT. Once planned, the clinician validates its design. Personalized manufacturing is carried out for the patient, which adapts without the need for manipulation during surgery.

## 3. Case Report and Results

A CAD-CAM planning of the titanium mesh (Avinent^®^, Santpedor, Spain) fixed with four microscrews (Avinent^®^, Santpedor, Spain) is performed. (Figure 2). The procedure is explained to the patient, she signs the informed consent, and the surgery is planned.

i.First surgery: Local anesthesia Articaine 4% with epinephrine (1:100,000) was administered. An intrasulcular incision was made from the mesial of 3.3 to the distal of 3.7, continuing along the anterior border of the external ramus of the mandible. Periosteal incisions were made to mobilize the vestibular flap and the upper fibers of the mylohyoid muscle were disinserted to passivate the lingual flap and thus obtain a tension-free closure (Figure 3). We then checked the position of the mesh and proceeded to collect bone from the ascending branch using the bone scraper (Micross^®^, Selecdent, Barcelona, Spain) (Figure 4A). Cortical perforations were made to promote bleeding (Figure 4B). We mixed the autologous bone with the xenograft (Tioss^®^, Sanhigia, Bujaraloz, Spain) in a 70:30 ratio, inserted it into the mesh, and the mesh was placed in the defect.

A 15 × 30 mm pericardial membrane (Lyoplant^®^, Sanhigia, Bujaraloz, Spain) was placed inside the titanium mesh. We fixed the mesh with two microscrews per buccal and two per lingual (Figure 5). Once in place, a 30 × 40 mm resorbable collagen membrane (Geistlich Bio-Guide^®^, Inibsa, Barcelona, Spain) was added to promote attachment of the periosteum to the resorbable membrane. We performed the closure using horizontal mattress stitches with non-absorbable monofilament polyamide suture (Supramyd 5/0) to evert the surgical edges and closed the wound with simple stitches (Figure 6). A post-operative orthopantomography was taken after surgery (Figure 7). Post-operative guidelines were prescribed; antibiotic and anti-inflammatory (Amoxicillin 750 mg 1 every 8 h for 7 days, Ibuprofen 600 mg 1 every 8 h as needed, and Paracetamol 1 g 4 h after ibuprofen if there is pain greater than 7 on the VAS scale). The use of 0.12% Chlorhexidine mouthwashes twice a day 24 h after the intervention was also recommended. During the first 15 days, we recommend a soft diet.

ii.The sutures were removed on day 21. A panoramic radiograph was taken after surgery. (Figure 8). Periodic controls were scheduled; every week during the first 2 months, every two weeks in the third and fourth months, and once a month up to 6 months.

No complications or wound dehiscence was observed until the sixth month when a small 2 mm dehiscence was observed in the distal area of the crest near tooth 3.7 (Figure 9).

In the sixth month, a CBCT and the future planning of the implants were scheduled.

iii.On the day of surgery, the titanium mesh and the microscrews were removed (Figure 9). When the mesh was removed, a soft consistency was observed in the most coronally newly formed bone and it was decided to postpone the placement of the implants and allow it to ossify for another month and a half. A panoramic radiograph was taken after surgery (Figure 10).iv.After 7 and a half months, the Avinent^®^ 3.8 × 8.5 implants were placed in position 3.5 and 4 × 8.5 in position 3.6 (Figure 11). The torque of the implants was greater than 45 N/cm. The ISQ of both implants was taken, being 82 buccal and palatal for the implant in position 3.5 and an ISQ of 57 buccal and palatal for the implant in position 3.6. The bone gain obtained was 1.84 and 1.92 mm in width and 4.2 and 3.78 mm in height for positions 3.5 and 3.6. Simultaneously with the placement of the implants, a bone biopsy was performed between the implants, using a 2 mm bone trephine (Sanhigia, Bujaraloz, Spain) (Figure 12). Three months after the placement of the implants, the implants were rehabilitated using metal-ceramic screw-retained crowns.

## 4. Discussion

Titanium mesh is a non-resorbable membrane that has been widely used for its mechanical properties for bone graft stabilization. The stiffness provides stability that is necessary to maintain the volume of the bone graft during wound healing [21]. Therefore, titanium meshes are valid for both horizontal and vertical regeneration. In the article published by Cucchi et al. [22], the authors showed that there are no statistically significant differences in vertical gain and the number of complications between titanium mesh and non-resorbable PTFE membranes. Instead, the article by Kim et al. [23] reports less infection in the bone graft if there is exposure to the titanium mesh since the dense surface of the mesh makes it more susceptible to bacterial adhesion. They also attribute it to the fact that soft tissue adhesion is easier thanks to the pores of the titanium meshes compared to PTFE membranes. Regarding the complications of titanium mesh, the most frequent complication is its exposure. The incidence of exposure varies depending on the articles from 20% to 66% [24,25]. These exposures have been divided into early exposures (during the first 4 months of regeneration) and late exposures (after 4 months of regeneration). If the titanium mesh is exposed during the first 4 months, it is recommended that the mesh be removed as soon as possible. If the exposure is performed after 4 months, it could cause resorption of 15–25% of the exposed bone graft [26]. In our case, it was exposed at the sixth month of regeneration and topical treatment with 0.2% topical Chlorhexidine two times a day was carried out until the day of mesh removal. According to the article by Corinaldesi et al. [27], when the titanium mesh is exposed and if the biomaterial is stabilized, regeneration can be guaranteed and superinfection can be prevented thanks to the mesh pores that play a critical role in maintaining blood flow and allowing hygiene. Therefore, compared to other regeneration techniques with a non-resorbable membrane of e-PTFE or d-PTFE, if there is an exposure to it, it is easier for graft infection to occur. The fabrication of the mesh corresponds to type 2, proposed by Shi et al. [19], which allows custom manufacturing for each patient based on 3D planning. This allows a precise adaptation of the mesh, without the need for intraoperative adaptations, which limits surgical time and postoperative complications. The clinical advantages and disadvantages of customized titanium membranes are clearly reviewed in the recent work by Shi et al. [19] and the authors recommend that research should be aimed at increasing the antibacterial and osteogenic capacity of the mesh, with the aim that the bone tissue growth is as expected.

On the other hand, given that titanium membranes require a second surgery, research is currently heading towards biodegradable materials with promising results. These membranes have increasingly better biocompatibility, good mechanical properties to maintain space, excellent properties to promote osteogenesis and reduce the risk of exposition [28].

Regarding the proportion of bone graft material, in our case we used xenograft and autologous particulate bone scraped from the mandibular ramus in a 30:70 ratio. This combination has been used in several studies [29]. The ratio used varies from 50:50 to 30:70 [24,30]. A bone xenograft is one of the most widely used bone graft biomaterials due to its high biocompatibility and slow resorption to maintain volume [6].

When performing the histological study, bone with a well-configured neoformed appearance and abundant lamellar structure is observed. It has been shown that autologous bone provides more vital bone in regeneration, although greater bone resorption. Additionally, in the biopsies performed in the literature, it can be seen that both the allograft and the autologous bone provide greater mimicry with the recipient’s bone. Regarding the xenograft, bone growth was observed around the xenograft particles, but in turn, some xenograft particle was observed that persisted [31,32].

Regarding bone gain with titanium mesh, studies show bone gains in width of up to 5 mm and vertical gains of up to 7 mm [19,33,34]. In our case, the bone gain obtained is lower than that shown in the literature. The horizontal gain was 1.88 ± 1 mm and the vertical gain was 4 ± 1 mm. The bone gain obtained is within that expected based on the planification and it has allowed us to place implants of 38 and 4 by 8.5, maintaining an adequate root–crown ratio for the case that we present.

To prevent soft tissue invagination through the pores of the titanium mesh, we use a resorbable collagen membrane. However, the study by Lim et al. [35], found no statistically significant differences in the exposure rate of titanium mesh with or without being covered with a resorbable collagen membrane, concluding that the collagen membrane does not reduce exposure or excessive soft tissue formation. 

Conventional titanium meshes can be performed intraoperatively, but the disadvantages are that shaping and bending of the mesh can lead to degradation of the titanium mesh, leading to sharp edges and early wound dehiscence. Today, thanks to advances in digitization, titanium meshes are manufactured using CAD-CAM technology. This advance provides a significantly shorter operating time since they adapt perfectly to the defect and do not require intraoperative shaping. They also have the most rounded edges and prevent tissue exposure around the edges. In turn, they require fewer fixation screws [36,37].

## 5. Conclusions

This case report and the recent literature on the matter help us to confirm that for vertical and horizontal defect reconstruction, guided bone regeneration using customized titanium mesh is a predictable technique due to its mechanical properties. With digitalization, clinical procedures using titanium mesh have improved by shortening surgery time and reducing trauma to the patient, which could increase the success rate of bone augmentation.

## Figures and Tables

**Figure 1 materials-15-06271-f001:**
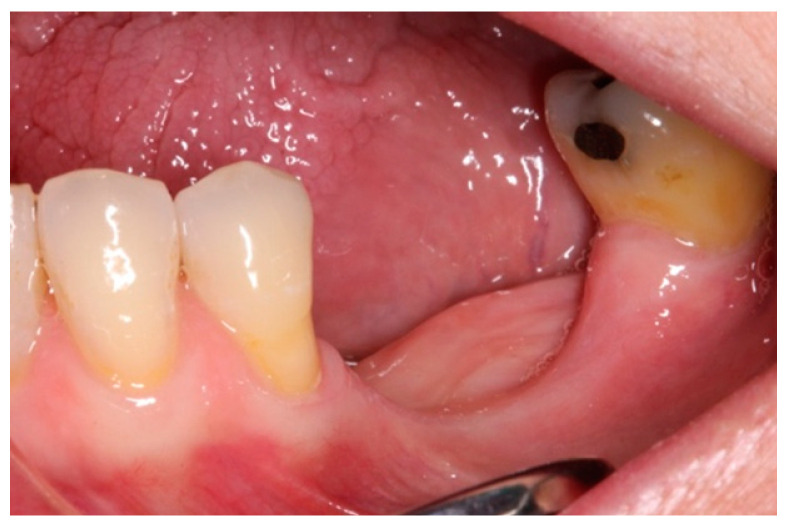
Vertical bone defect in the intraoral examination of the third quadrant.

**Figure 2 materials-15-06271-f002:**

(**A**,**B**) Third quadrant CBCT measurements. (**C**) 3D mesh planning.

**Figure 3 materials-15-06271-f003:**
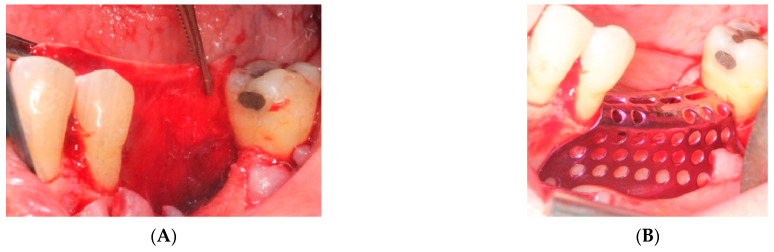
(**A**) Passivation of the lingual flap. (**B**) Checking the titanium mesh.

**Figure 4 materials-15-06271-f004:**
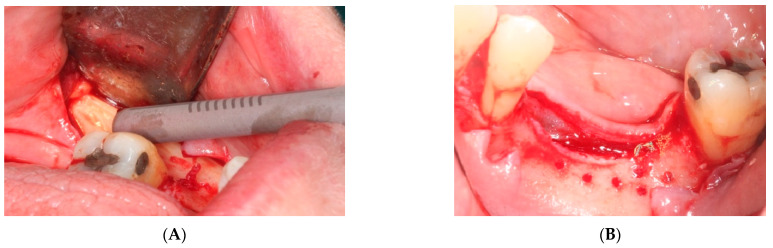
(**A**) Obtaining autologous bone from the mandibular ramus. (**B**) Perforations in the mandibular bone with a #6 tungsten carbide bur.

**Figure 5 materials-15-06271-f005:**
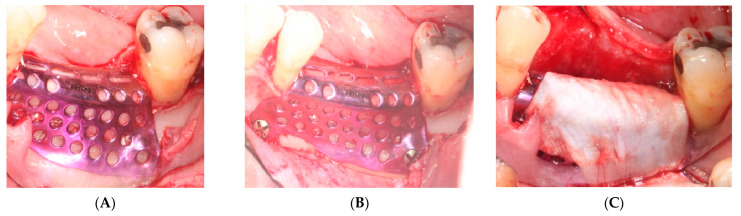
(**A**,**B**) Placement of the mesh with the biomaterial inside. (**C**) Placement of the collagen membrane outside the titanium mesh.

**Figure 6 materials-15-06271-f006:**
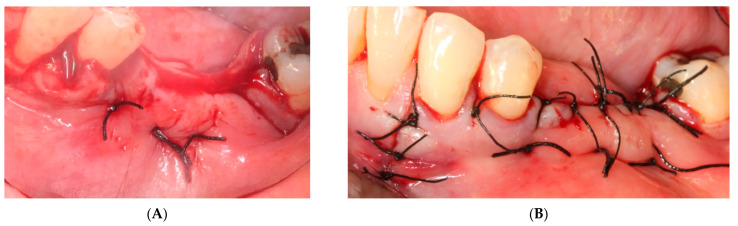
(**A**) Horizontal mattresses to evert the edges. (**B**) Simple Interrupted Suture.

**Figure 7 materials-15-06271-f007:**
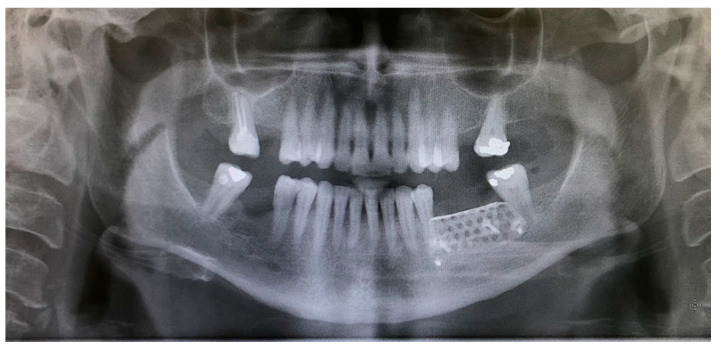
Post-operative orthopantomography.

**Figure 8 materials-15-06271-f008:**
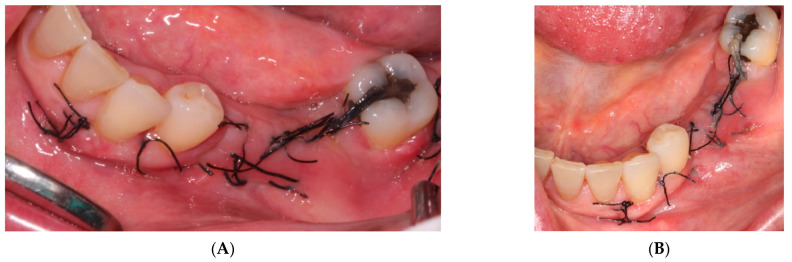
(**A**) Control at 7 days. (**B**) Control at 14 days, a good appearance of the wound is observed.

**Figure 9 materials-15-06271-f009:**
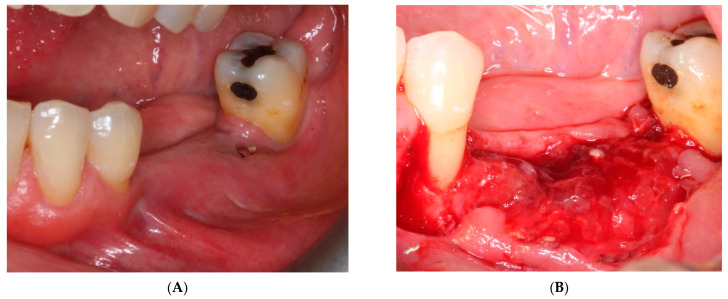
(**A**) Control at 6 months and discreet exposure of the titanium mesh. (**B**) Mesh removal. Immature bone is observed, clear gain is seen in the two areas near the bone peaks and less loss in the central area of the defect.

**Figure 10 materials-15-06271-f010:**
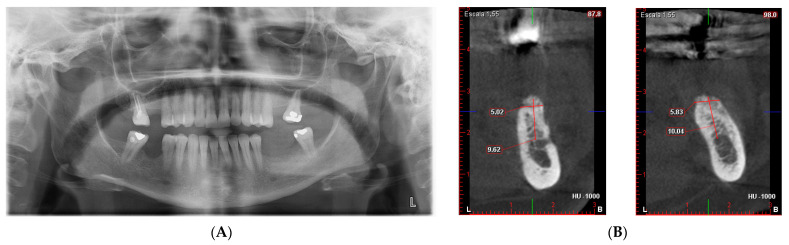
(**A**) Orthopantomography after mesh removal. (**B**) CBCT with measurements for implant placement.

**Figure 11 materials-15-06271-f011:**
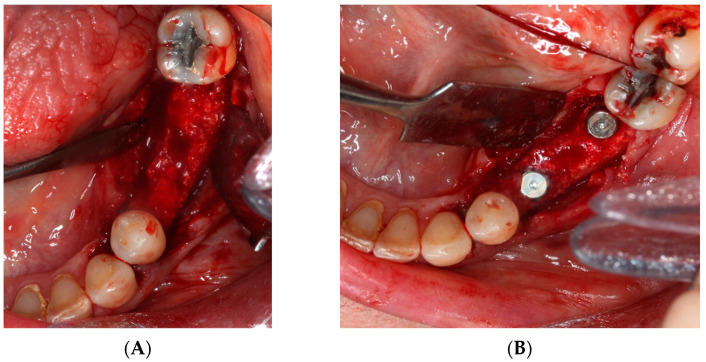
(**A**) Implant placement surgery (**B**) obtaining the bone trephine.

**Figure 12 materials-15-06271-f012:**
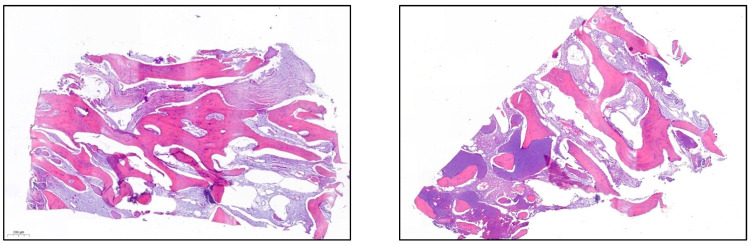
A well-formed and mature bone structure is observed.

## Data Availability

Not applicable.

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
