# Peer review of "Customized Titanium Mesh for Guided Bone Regeneration with Autologous Bone and Xenograft"

_materials, 2022, doi:10.3390/ma15186271_

Round 1

Reviewer 1 Report

The manuscript "Customized titanium mesh for guided bone regeneration with autologous bone and xenograft" presents some interesting results. However, the main audience of "Materials" consists of materials researchers. Therefore, I suggest that the authors include more informations about the titanium mesh used in the investigation. Even the mesh being a commercial product, some basic informations can be obtaind from the manufacturer. 

Reviewer 2 Report

The manuscript titled “Customized titanium mesh for guided bone regeneration with 2 autologous bone and xenograft” by Faus et al. reports guided bone regeneration for horizontal and vertical bone defects. A detailed in vivo evaluation of a 59-year-old female has been carried out. While the overall work is interesting, it suffers several drawbacks which can most probably prevent it from attracting a wide readership. Hence, the following queries/questions need to be addressed before the manuscript can be considered suitable for publication:

1.     The abstract must be thoroughly improved. The authors mention the use of customized titanium mesh in the title, while in the abstract, there is no mention of the customization. The abstract must reflect the whole work plan and important results. The readers read the abstract to decide whether it is worth reading the article. Hence, it must be improved.

2.     The introduction is very short and weak. The authors are unable to effectively discuss the details of the background study and the drawbacks that this work will address. Further, the importance and novelty of the current study are not mentioned clearly.

3.     The CAD-CAM planning is customized and it should be mentioned in more detail. This information must be part of the materials and methods section.

4.     One comparative section/table about the results of existing implants to address similar problems i.e. guided bone regeneration must be included in this manuscript.

5.     The conclusion section is weak. Essential details about the obtained results are missing. It can be made stronger by adding some quantitative results about bone volume.

6.     The recent related work is not cited, wherever applicable please include a recent citation.

Reviewer 3 Report

Your research paper is very interesting. However, revisions to the study design are required for publication in this journal. This research paper is a case report. Please demonstrate the effectiveness of posterior bone augmentation using titanium mesh using multiple cases and using statistical methods.

Round 2

Reviewer 2 Report

The revised manuscript is improved significantly and it can be published as a case report.

Reviewer 3 Report

This study is a case report and is not suitable considering the journal level.

Please increase the number of cases and reconsider
